# Non-destructive multi-sensor core logging allows rapid imaging and estimation of frozen bulk density and volumetric ice content in permafrost cores

Joel Pumple [1, *], Alistair Monteath [1], Jordan Harvey [1], Mahya Roustaei [1, 3], Alejandro Alvarez [1], Casey Buchanan [1, 2] and Duane Froese [1, *]

[1] Department of Earth and Atmospheric Sciences, University of Alberta, Edmonton, Canada
[2] Yukon University, Whitehorse, YT
[3] Department of Civil Engineering, Geotechnics Laboratory, Ghent University, Technologiepark, 68, 9052 Zwijnaarde, Ghent, Belgium

Correspondence to: Joel Pumple (*pumple@ualberta.ca*) or Duane Froese (duane.froese@ualberta.ca)

**Abstract.** Permafrost cores provide physical samples that can be used to measure the characteristics of frozen ground. Measurement of core physical properties, however, are typically destructive and time intensive. In this study, multi-sensor core logging (MSCL) is used to provide a rapid (~2-3 cm core depth per minute), high-resolution, non-destructive method to image and collect the physical properties of permafrost cores, allowing the visualization of cryostructures, estimation of frozen bulk density, magnetic susceptibility and volumetric ice content. Six permafrost cores with differing properties were analyzed using MSCL and compared with established destructive analyses to assess the potential of this instrument both in terms of accuracy and relative rate of data acquisition. A calibration procedure is presented for gamma ray attenuation from a $^{137}$Cs source that is specific to frozen core materials. This accurately estimates frozen bulk density over the wide range of material densities found in permafrost. MSCL frozen bulk density data show agreement with destructive analyses based on discrete sample measurements, RMSE = 0.067 g/cm$^3$. Frozen bulk density data from the gamma attenuation, along with soil dry bulk density measurements for different sediment types, are used to estimate volumetric ice content. This approach does require an estimation of the soil dry bulk density and assumption of air content. However, the averaged results for this method show good agreement with an RMSE = 6.7%., illustrating MSCL can provide non-destructive estimates of volumetric ice contents and a digital archive of permafrost cores for future applications.

## 1 Introduction

Permafrost cores are typically collected from remote locations at great expense. Despite the considerable cost involved in the recovery, transportation and storage of permafrost cores most analytical methods are destructive and rarely preserve physical or digital archives for future work. Typically, permafrost samples are qualitatively described in the field based on visible or nonvisible ice content (Pihlainen and Johnson, 1963; Linell and Kaplar, 1966; ASTM D4083-89 2016; CAN/BNQ 2501-500, 2017). These descriptions may include cryostructures of the ice and sediment, and a subset of these observations may be combined with destructive analyses to quantify volumetric ice content, excess ice, gravimetric moisture content and soil organic matter (Pihlainen and Johnston, 1963; Andersland and Ladanyi, 1994; Kokelj and Burn, 2003; French and Shur, 2010; Stephani et al., 2010). These properties can be used to estimate thaw potential or for the interpretation of cryostratigraphy. The accurate measurement of these variables is important for designing climate-resilient infrastructure and predicting the impacts of permafrost thaw. A second concern with permafrost characterization arises because increasing air temperatures and thawing permafrost mean that permafrost landscapes are rapidly changing. This transformation of permafrost regions emphasizes the need to archive material that may not be available for future research (Jorgenson et al., 2010; Throop et al., 2012), or may be appropriate for complementary applications.

In recent decades there have been advances in using laboratory based non-destructive methods for measuring the physical properties of frozen materials, including gamma attenuation (e.g. Zhou et al., 2014; Lenz et al., 2015), ultrasonic methods (Liu et al., 2023), nuclear magnetic resonance (Kleinberg and Griffin, 2005; Kruse and Darrow, 2017) and computed tomography scanning (e.g. Calmels and Allard, 2004; Carbonneau et al., 2011; Darrow and Lieblappen, 2020). This study investigates the use of multi-sensor core logging (MSCL) to characterize permafrost cores. The MSCL is well established in its application for marine sediments (Weber et al., 1997; Gunn and Best, 1998), landslide assessments (Hunt et al., 2011; Vardy et al., 2012), contaminated sediments (Kuras et al., 2016) and environmental studies of lake sediments (Smol et al., 2001; Fortin et al., 2013). However, to our knowledge, the application of MSCL on frozen materials has not been developed. Here, we use MSCL systems to measure the physical properties of frozen cores with gamma ray attenuation, non-contact resistivity, magnetic susceptibility, and high-resolution line-scan imagery. The aims of this paper are to (1) develop calibration methods for gamma attenuation density determination based on physical principles that are specific to frozen materials. (2) Analyze six permafrost cores with differing physical properties to test the suitability of MSCL for non-destructive characterization. (3) Finally, we will compare these results with destructive measurements at similar spatial scales on discrete sediment samples to test the accuracy of the non-destructive MSCL method.

## 2 Methods and materials

### 2.1 Multi Sensor Core Logger (MSCL)

This study uses a Geotek (U.K.) MSCL in the Permafrost ArChives Science (PACS) Laboratory at the University of Alberta, Canada. The MSCL is a floor-mounted, automated logging system that passes core along a track, either below or between analytical sensors (Fig. 1). This system can be used to analyze whole or split cores up to 150 cm long, with outer diameters less than 15 cm. The PACS Laboratory MSCL is equipped with two magnetic susceptibility instruments (Bartington Loop sensor for whole cores and a contact magnetic susceptibility sensor for split core), a high-resolution line-scan camera, a

Caesium-137 ($^{137}$Cs) gamma ray source and detector and a non-contact resistivity sensor. In this study, we used the MSCL split core configuration. The core boat used for both the cores and calibration pieces is a 10 cm diameter PVC pipe with cores sitting in an open style half pipe and the calibration piece in a whole pipe with a viewing window to allow for submersion of the calibration piece within water/ice (Fig. 1).

The MSCL system within the PACS Laboratory is located in a space kept at 23 °C. Although the PACS Laboratory

has cold rooms and walk-in freezer spaces, in the design phase, the decision was made to develop a core scanning method that could be completed at room temperature. This provided savings in construction costs and is likely to reduce maintenance time/costs compared with a cold room installation where higher moisture and condensation could impact the electronics and camera. As a result, we developed a method for working on frozen cores that would maintain the integrity of the cores throughout analysis at room temperature.

### 2.1.1 Gamma Attenuation Operating Principles

The MSCL is equipped with a Caesium-137 ($^{137}$Cs) gamma ray source and detector for measuring the attenuation of gamma rays through the core. These measurements are also referred to as gamma ray attenuation porosity evaluator (GRAPE) density measurements (Evans, 1965), and are commonly used in marine and lacustrine sediment cores (Weber et al., 1997; Geotek, 2021). In its simplest form, a gamma ray source passes a beam of collimated gamma rays between a 2.5 or 5 mm aperture and

through the centre of the core. Photons passing through the core are detected at the sensor, while some are attenuated, largely by Compton scattering, resulting in partial energy loss. When other variables, including the gamma source intensity, Compton attenuation coefficient and core thickness are known, the degree of energy loss can be used to calculate bulk sediment density (Geotek, 2021). As the density of ice (0.9 g/cm$^3$) is substantially lower than minerals that typically make up sediment cores (e.g., quartz 2.65 g/cm$^3$), density changes in ice-rich permafrost cores are principally driven by sediment porosity/organic

content and the abundance of ice within the sediments.

### 2.1.2 PACS Laboratory Permafrost Core Operating Protocol

The methods described here have been developed to provide accurate core frozen bulk density measurements while mitigating the risk of thaw during analysis. These include experiments designed to identify suitable gamma ray attenuation analytical times while minimizing core exposure at temperatures >0 °C.

Accurate gamma ray attenuation measurements require a minimum of approximately 100,000 counts per measurement at the gamma ray detector (Geotek per comms, 2022). This is achieved through longer measurement times (up to 30 seconds) for calibration. To assess the minimum count time required to meet this threshold three permafrost cores with different sediment properties and ice contents were analyzed using 30 second, 10 second, 5 second and two second count times. The 5 mm aperture was used to maximize counts per second. Our results showed that a five second count time produced consistent results (>100,000 counts per measurement) that agreed well with longer count times (10 second and 30 second) and were adopted for subsequent measurements. This approach resulted in an average acquisition time of 2-3 cm/minute using a 5 mm aperture and 5 mm sampling resolution. After removal from a -25 °C freezer, the internal temperature of a test half core increased to ~ -12 °C during a 30-minute acquisition. This approach met the requirements to accurately measure sediment densities of our samples. Denser or thicker cores would require longer analytical time for similar counts. As well as this, count times required to reach minimum counts at the detector will vary over time as the Caesium-137 source decays. The data collected in this study are under colder temperatures than ambient field conditions. Future development will focus on designing of a chilling boat for the samples to maintain samples at much warmer temperatures (-0.5 – 5 C°) during measurement.

### 2.1.3 Development of Calibration Standards for Frozen Materials

As gamma bulk density, to our knowledge, has not previously been used to analyze permafrost cores via MSCL, several experiments were completed to develop a calibration protocol. To calculate frozen bulk density from gamma ray attenuation the MSCL uses a stepped aluminum calibration block of known thickness and density. This is analyzed using a fixed count time (typically 30 seconds) at the gamma detector and runs across the length of the calibration block, including a small section outside the calibration block to capture the core casing/core boat and background material. In common practice, the background material is air for dry cores and water for saturated cores. The density and thickness of the aluminum block, along with the known density of the background (air, water, or ice as we discuss), are then used to convert the resulting counts per second into $g/cm^3$. In our initial experiments, we used the default aluminum calibration block from the manufacturer; however, due to the length of the steps (3 cm), the determination of the average counts between steps was difficult to consistently capture using the shorter count times. These inconsistencies were further affected by stainless steel screws that penetrated either end of the aluminum block. To meet these challenges, we constructed a new calibration block based on the manufacturer's design, but with longer steps (6 cm rather than 3 cm) and no screws (Fig. 2).

Permafrost samples typically include variable ice content and minimal free water below ~ -10 ºC. The calibration approach outlined by the manufacturer was not designed for permafrost applications it did not span the range of densities and

materials encountered in the frozen sediments (ice, 0.9 g/cm$^3$ to clast-rich diamict, >2.4 g/cm$^3$). When analyzing unfrozen sediment cores, calibration of bulk density typically uses the aluminum calibration piece submerged in water as pore-spaces are assumed to be saturated. As water is denser than ice, this leads to a potential source of bias in measuring the density of ice-rich materials (e.g., peat and ice lenses), which lie outside of the calibration range. To extend the calibration range, and account for the density differences between ice and water, the newly developed stepped standard piece was submerged in water and frozen prior to calibration (Fig. 2). This frozen calibration piece produced results more comparable to frozen materials of known density (e.g., pure ice and peat that are expected to have densities ~0.9 g/cm$^3$). Unsurprisingly, the offset between frozen and unfrozen calibration was greatest in ice-rich materials. Figure 2 shows a piece of foam on the left side of the calibration piece (to keep it level) and highlights the depth of the water/ice surrounding the calibration piece (blue dashed line). The foam piece results in a local density low or high depending on the surrounding medium. It should also be noted that the accuracy of the calibration when using this frozen calibration piece was sensitive to the depth of the surrounding ice. The aim was to have the ice level with the flat surface of the calibration piece to minimize variability. Repeat analyses were completed on frozen half cores with a 5 second count time and 5 mm gamma source aperture to quantify uncertainty in the density measurement. The measured uncertainty based on 15 repeat measurements is ± 0.05 g/cm$^3$.

Beam spreading below the gamma source aperture results in the beam diameter increasing from 5 mm at the aperture to ~ 10 mm at the core surface, and likely expands further to ~15 mm at the detector. This cone shape causes an edge effect that leads to bulk density measurements decreasing at or near any core breaks or sharp density contrasts (eg. top and bottom of a core or the calibration piece; Figure 2). The gamma attenuation data from the bottom and top of the cores has been edited to remove edge affected data.

### 2.1.4 Magnetic Susceptibility (Mag Sus)

The MSCL is equipped with two magnetic susceptibility instruments: a Bartington MS2E point sensor and a Bartington MS2C loop sensor (160 mm diameter). In this paper, the results from the Bartington MS2E point sensor are used as the measurement field is finer (approximately 2 cm) than the Bartington MS2C loop sensor (approximately 7 cm). This resolution increases the likelihood of capturing sharp cryostratigraphic transitions (e.g., ice lenses). When analysing core materials, the instruments were zeroed and calibrated against instrument specific Bartington standards before and following the analyses. Bartington MS2E point sensor measurements follow the same central vertical transect as the gamma density measurements allowing comparison between the two datasets.

### 2.1.5 Non-Contact Resistivity (NCR)

The Non-Contact Resistivity (NCR) sensor induces a high frequency magnetic field in the core that in turn induces electrical currents which are inversely proportional to resistivity. These are measured using receiver coils and compared with identical receiver coils operating in the air (Geotek, 2021). In permafrost cores, these electrical currents are likely to be altered by the differing abundance of ice and water because of the variable conductivities of these two materials.

The NCR instrument was tested on frozen materials at variable temperatures (-5 to -25 C°), but was strongly affected by the temperature of the cores and attempts to analyze frozen cores were hampered by substantial analytical drift between measurements. We recognize that unfrozen water content will be minimal at temperature below -5 C° and so an alternative insulated core boat would be needed if the sensors temperature sensitivity could be addressed. Similar problems were encountered during studies of Antarctic Ocean cores by Niessen et al. (2007), who did not consider the majority of their NCR

results reliable. An attempt was made to mitigate this issue by 'cooling' the sensor using a synthetic frozen core, however, drift during the analyses was considered too great. These issues meant that despite the considerable potential for measuring unfrozen water content in permafrost cores using NCR, at present, the temperature sensitivity of the instrument makes it unsuitable for this application. The NCR sensor is not discussed further because of this limitation.

### 2.1.6 Line-scan Imagery

Split cores were photographed using the line-scan camera equipped on the MSCL. The Geotek camera uses a Canon 50 mm lens focused through a thin viewing window in the lighting box. The resolution of the resulting images ranges from 100-micron pixels (low resolution setting) to 25 micron pixels (high resolution setting) and all images are exported as 48-bit RGB TIFF files. Moisture scavenging from the room caused some cores to build up ice on the core surfaces when scanned at ≤-20 °C, despite the low humidity conditions which are common in Alberta, Canada. This led to overexposure and poor image quality

in some sections. To mitigate this problem cores were warmed to -5 °C before imaging which improved image quality in both low- and high-resolution scans. This meant that core images and physical properties were measured in separate core runs, to minimize sample thaw potential during analyses.

### 2.2 Physical Density Measurements

To independently assess density measurements made using gamma attenuation, both volumetric and gravimetric methods were

completed on the same cores. The cores were subsampled as close as possible to the transect analyzed by the MSCL gamma density instrument (Fig. 3).

Destructive analyses for frozen bulk density, volumetric and excess ice contents follow methods described by Kokelj and Burn (2003), but are modified to provide higher resolution for comparison with the MSCL data. Instead of using whole (10 cm) core segments we used the cuboid method as described by Bandara et al. (2019), which is based on precisely measured

cuboids sampled throughout core segments. The cuboids in this study were processed in a walk-in freezer held at -7 °C. The first step was to remove material from the outer core edges that may have been thawed during coring or affected by sample storage. Core segments were split lengthwise using a rock saw equipped with a 35 cm diameter diamond cutting wheel (Fig. 4A). Cuboid aliquots were cut from half of the split core while the other half was retained as an archive (Fig. 4A). The rounded edges were removed from the half core to reveal an internal slab (Fig. 4B). For this study a duplicate set of cuboids were

collected by cutting the internal slab in half (Fig. 4C). Approximately 3 cm$^3$ aliquots were subsampled from the cores to ensure that cuboids did not fracture or disintegrate during sampling because of the lower ice content (Fig. 4D). Digital calipers (±0.01

mm) and a digital analytical balance (±0.01 g precision) were used to measure physical dimensions and mass, respectively, to calculate the frozen bulk density (Fig. 4E and 4F). The cuboids were then thawed at room temperature for 24 hours in glass beakers covered in parafilm to minimize evaporative loss (Fig. 4G & 4H). Any excess moisture was removed from the beakers containing the thawed samples and the sample weight was recorded again to calculate excess moisture content (Fig. 4I). The cuboids were then dried in an oven for 24 hours at 105 °C and reweighed to calculate both volumetric ice content and gravimetric moisture content (Fig. 4J and 4K). Finally, the remaining dried material was heated at 550 °C for four hours to determine the percent organic content via loss on ignition (LOI) (Fig. 4L; Heiri et al., 2001). The cuboid method provides an opportunity to collect pH and conductivity measurements from ice rich samples following the thawing stage, however, for this study these data were not collected. We recognize the importance of salinity in thaw sensitive permafrost regions, however, given the analytical constraints, thermal stability was top priority during our analysis. The aim is to consider free water and salinity in future studies using alternative non-destructive methods (e.g., Roustaei et al., 2022).

## 2.3 Volumetric Ice Content Estimates

We estimate the volumetric ice content of cores from the frozen bulk density following the approach used by Lin et al. (2020). This approach relates volumetric ice content and frozen bulk density as:

$$\rho = \frac{m}{V} = \frac{m_s + m_i + m_w + m_a}{V} = \frac{\rho_s v_s + \rho_i v_i + \rho_w v_w + \rho_a v_a}{V} \tag{1}$$

Where $\rho$ is density (g/cm$^3$), m is mass (g), and V is volume (cm$^3$). Subscripts *s, i, w,* and *a* represent dry soil content, volumetric ice content (both segregated ice and pore ice), unfrozen water content, and air content, respectively. V is the volume of the permafrost sample and is assumed as one unit volume so the above simplified to:

$$\rho = \rho_s v_s + \rho_i v_i + \rho_w v_w + \rho_a v_a \tag{2}$$

And by considering the mean volumetric content of unfrozen water and air equal to 1 and 2%, respectively, as well as 0.9 and 1 g/cm$^3$ for the density of ice and water, and ignoring the density of air, the equation is simplified to:

$$\rho = 2.04 v_s + 0.9 v_i + 0.02 \tag{3}$$

Lin et al. (2020) uses a value of 2.04 g/cm$^3$ for the soil dry bulk density of homogeneous permafrost samples from the Qinghai-Tibet Plateau, while in this study, soil dry bulk densities for each core are measured using the cuboid method, with most ranging from 1.4 to 2.6 g/cm$^3$ (Table 1). The largest sources of uncertainty using this approach are the estimated soil dry bulk density value and assumed 2% air content for all cores apart from the peat core and top of the transition core. Organic content being the most important variable controlling both the overall soil dry bulk density and sample air content we used an average soil dry bulk density for peat collected from Kazemian et al. (2011) and Motorin et al. (2017) of 1.55 g/cm$^3$ for the peat core and 1.75 g/cm$^3$ for the top of the transition core as it has lower organic content and higher mineral content. When we compared these values with the cuboid method VIC, these soil dry bulk densities relate to an average air content of up to ~6%.

Although this method requires the estimation of both the dry soil bulk density and sample air content it represents an important non-destructive tool for the permafrost core-based sciences. We recognize the MSCL can only analyse core style materials, however, we present this method as an alternative which represents both a cost and time savings relative to other methods for non-destructive the extraction of volumetric ice content such as nuclear magnetic resonance and computed tomography scanning.

## 2.4 Materials

Six permafrost cores that represent common materials encountered in permafrost regions were analyzed in this study. Each core segment was selected based on relatively simple cryostructures to minimize lateral heterogeneity (Table 1).

## 3 Results

### 3.1 Ice-wedge (BH19-204)

This core consists of sediment-poor ground ice with subvertical foliations. This represents a typical ice-wedge core from permafrost affected regions (Murton and French, 1994). The ice-wedge core was scanned on the MSCL for frozen bulk density but was not subsampled for cuboid measurements due to the high ice volume and lack of sediment in the core (Fig. 5). The mean density based on gamma attenuation is $0.90 \pm 0.03$ g/cm$^3$, while the average, minimum and maximum magnetic susceptibility values for this core are -1.55 SI x $10^{-5}$, -3.64 SI x $10^{-5}$, and 2.38 SI x $10^{-5}$, respectively.

### 3.2 Ice-rich Silt (BH18-211)

This core consists of ice-rich, organic sandy silt dominated by a wavy-micro-lenticular cryostructure and a single ice layer ~1 cm thick. Both MSCL and cuboid data were collected from the core (Fig. 6). The average density for the gamma attenuation and cuboid method are $1.27 \pm 0.1$ g/cm$^3$ and $1.24 \pm 0.1$ g/cm$^3$, respectively, and are consistent through the core. The volumetric ice content for the two datasets shows strong agreement apart from the ice-poor regions in the core where the MSCL method estimates lower values relative to the cuboid data. The average, minimum, and maximum magnetic susceptibility values for this core are 13.44 SI x $10^{-5}$, 0.79 SI x $10^{-5}$, and 29.23 SI x $10^{-5}$, respectively.

### 3.3 Transition Core (BH12F-138)

This core contains a sharp transition between an overlying ice-rich silty peat and an underlying ice-poor inorganic silt with micro-lenticular cryostructures. Frozen bulk density from the gamma attenuation and cuboid methods in the top half of the core (ice-rich silty peat) are $0.98 \pm 0.02$ g/cm$^3$ and $1.02 \pm 0.03$ g/cm$^3$, respectively (Fig. 7). The mean density for the gamma attenuation and cuboid method in the bottom half of the core (ice-poor inorganic silt) are $1.31 \pm 0.14$ g/cm$^3$ and $1.32 \pm 0.25$ g/cm$^3$, respectively. Both the gamma attenuation and cuboid frozen bulk density data resolve the transition. The two datasets

track well except for the high-density peak (volumetric ice content low) in the lower half (below 8 cm depth) of the core which is missed by the 2 cm resolution cuboid data. The average, minimum, and maximum magnetic susceptibility values for this core are 32.08 SI x $10^{-5}$, -0.66 SI x $10^{-5}$, and 151.64 SI x $10^{-5}$, respectively.

### 3.4 Diamicton (BS19-3-6)

BS19-3-6 is an ice-rich diamicton with suspended and crustal cryostructures. The mean density for the gamma attenuation and cuboid methods are $1.45 \pm 0.11$ g/cm$^3$ and $1.46 \pm 0.11$ g/cm$^3$, respectively (Fig. 8). Overall, this core shows good agreement between the MSCL and cuboid methods. The datasets differ notably from 2-4 cm depth and 10-12 cm depth where the gamma attenuation dataset shifts toward lower density values. The cause of these differences is discussed in Section 4.1. The mean, minimum, and maximum magnetic susceptibility values for this core are 36.83 SI x $10^{-5}$, 6.42 SI x $10^{-5}$, and 230.34 SI x $10^{-5}$,

respectively.

### 3.5 Ice-poor Silt (BH20B-337)

This core consists of ice-poor sandy silt with a massive/structureless cryostructure (i.e., lacking visible ice). Silt in this core is relatively homogeneous, with the mean density for the gamma attenuation and cuboid method being $1.75 \pm 0.07$ g/cm$^3$ and $1.85 \pm 0.05$ g/cm$^3$, respectively (Fig. 9). The cuboid data shows minimal variability throughout the core; however, the higher-

resolution gamma attenuation data shows greater variability in frozen bulk density. Overall, the gamma attenuation frozen bulk density data agrees with the cuboid data. Similarly, the MSCL volumetric ice content agrees with the cuboid data. The mean, minimum, and maximum magnetic susceptibility values for this core are 354.14 SI x $10^{-5}$, 285.89 SI x $10^{-5}$, and 409.29 SI x $10^{-5}$, respectively. This core contains the highest magnetic susceptibility values of all the cores in the study.

### 3.6 Ice-rich Peat (DH13-589)

The peat core consists of homogenous organics with an organic-matrix cryostructure displaying some visible ice within pore spaces. The mean density for the gamma attenuation and cuboid method are $0.9 \pm 0.01$ g/cm$^3$ and $0.91 \pm 0.08$ g/cm$^3$, respectively (Fig. 10). The gamma attenuation data agree well with the cuboid measurements and highlight the ability of the gamma attenuation to capture subtle changes in the density of frozen peat. The mean, minimum, and maximum magnetic susceptibility values for this core are -1.43 SI x $10^{-5}$, -2.19 SI x $10^{-5}$, and -0.73 SI x $10^{-5}$, respectively.

### 4 Discussion

The non-destructive MSCL method provides a rapid method to collect core images, frozen bulk density, magnetic susceptibility and data to estimate volumetric ice contents from a variety of permafrost materials. Cores analyzed in this study were selected to represent the diverse physical properties and materials encountered in permafrost regions. With the exception of the diamicton core (BS19-3-6), the cores were selected based on their assumed low degree of lateral heterogeneity along the cores

central vertical axes representing the main sample locations. The root mean square error (RMSE) has been calculated for each discrete sample comparison and overall averaged whole core analysis. This statistical metric effectively illustrates the ability of the non-destructive method to collect similar results to the destructive method. Figure 11 shows strong agreement between the methods with the individual core based average RMSE values are as follows: ice-rich silt RMSE = 0.085 g/cm$^3$ , silty peat (top of transition core) RMSE = 0.062 g/cm$^3$, sandy silt (bottom of transition core) RMSE = 0.103 g/cm$^3$, diamicton RMSE = 0.064 g/cm$^3$, ice-poor silt RMSE = 0.051 g/cm$^3$, and ice-rich peat RMSE = 0.038 g/cm$^3$, and an overall average RMSE of 0.067 g/cm$^3$, illustrating the reliability of the MSCL method to non-destructively collect physical properties of permafrost cores.

In addition to frozen bulk density data, volumetric ice content data was measured using the cuboid method and estimated using gamma attenuation frozen bulk density following the methods first outlined by Lin et al. (2020). However, rather than a single value for the soil dry bulk density, we estimated the average soil dry bulk density for each core using the cuboid data with an assumed air content. These estimated values showed considerable variation (ca. 1.4-2.5 g/cm3), but with additional estimations of more permafrost types, reliable estimates for different sediment types can likely be established as well as expected air contents. An added challenge to reliable volumetric ice content estimates is the need for careful core preparation of frozen materials such that thickness is constant and core samples sit in contact with the core boat as discussed in Section 4.1. Overall, the gamma attenuation and cuboid volumetric ice content results show good agreement with a RMSE of 6.7% and demonstrate the potential for systematic and reliable estimation of volumetric ice content of permafrost cores non-destructively (Fig. 12). The individual core-based average RMSE values are as follows: ice-rich silt RMSE = 8.4%, silty peat (top of transition core) RMSE = 8.1%, sandy silt (bottom of transition core) RMSE = 7.5%, diamicton RMSE = 8.2%, ice-poor silt RMSE = 4.1%, and ice-rich peat RMSE = 3.9%.

Figure 13 shows a comparison between the destructive and non-destructive results for bulk density, displaying all individual sample results. Overall, the non-destructive method shows the ability to accurately recreate the results of the cuboid method despite the contrasting sample resolution and location. Figure 14 shows that increased core heterogeneity results in increased RMSE, in terms of volumetric ice content estimations. The ice-poor silt and peat cores which are the most homogenous in terms of both ice content and bulk density display the lowest RMSE. The higher RMSE values seen in the heterogenous samples can be related to the difference in sample resolution and location between the cuboid method and MSCL method. Additionally, the core shape issue with the MSCL thickness laser, as discussed in the next section, had a compounding impact on the volumetric ice content data.

## 4.1 Density Variability and Lateral Heterogeneity

Variations in measured densities of permafrost cores may reflect real heterogeneities in physical properties, or alternatively, artifacts introduced due to core preparation or mounting of frozen materials. For example, the diamicton core (BS19-3-6) has a strong lateral heterogeneity in composition that is highlighted by the offset between the cuboid sample locations and the central path of the MSCL gamma beam (Fig. 8). From 2-4 cm depth in the diamicton core, the MSCL dataset records a large

difference in frozen bulk density, and as a result, also in estimated volumetric ice content relative to the cuboid data. (Fig. 3).
These cores were selected in part because of their assumed lack of lateral heterogeneity to minimize the impact when

comparing the results between the destructive and non-destructive methods. The diamicton example highlights the importance
of that consideration. At this depth in the core a clast caused a local density high and a corresponding low in volumetric ice
content in the cuboid sample data. This high density and low volumetric ice content area was not observed in the gamma
attenuation data reflecting the lateral offset in data collection location along the core. This same clast also caused a local
magnetic susceptibility peak in this core, marking one of the few exceptions to the otherwise inverse relation between magnetic

susceptibility and ice content observed in the cores.

One challenge in using the MSCL for frozen cores occurs because frozen cores behave rigidly in the core boat unlike
unfrozen, saturated materials which tend to deform to fill the core boat. This can result in air gaps between the core and core
boat. Figure 15 illustrates the relations between the core boat and core as well as a profile of the diamicton core (BS19-3-6).
This core exhibits substantial rounding of the bottom that is not visible to the MSCL thickness laser, which only measures the

310 core surface, resulting in local overrepresentation of core thickness and thus density measurement inaccuracy. Similarly, but
less obvious, the ice-poor silt core appeared less dense because the upper ~5 cm of the core had a slight air gap between the
core and the core boat. Again, this gap is not detected by the laser measurements and results in an inaccurate measurement of
the frozen bulk density. This source of error underscores the importance of core preparation and positioning prior to scanning.
This issue has since been resolved further reducing the sources of error for this MSCL method.

Although the peat core had a relatively consistent shape, the underside of the core did show minor variations in
thickness. The impact of these differences is not immediately evident in the MSCL frozen bulk density measurements.
However, these differences are amplified in the volumetric ice content estimates. As seen in Figure 10, the volumetric ice
content data between 14-20 cm depth increases close to, and in some areas slightly above, 100%. These values coincide with
the lowest recorded frozen bulk densities in the profile and reflect the influence of small gaps between the core and the core

boat. These gaps would have less impact on a core with a higher overall frozen bulk density while they considerably affect
volumetric ice content estimations of ice-rich material with low frozen bulk density.

Another source of error in density measurements can occur at core breaks. The ice-rich organic silt core contains a
core break, which is reflected in the data (Fig. 6). At 10 cm depth, the core break results in a higher density peak and thus a
lower volumetric ice content estimate. The density peak associated with the core break occurs because the MSCL laser

thickness records a decrease in core thickness while the gamma ray spot size overlaps the core break. The slight decrease in
counts per second combined with a large decrease in core thickness at the core break compound to create an inference in the
model toward higher density, and thus lower volumetric ice content.

**4.2 Within Core Transitions**

Variations in grain size, ice content, and organic content can mark sharp changes in core cryostratigraphy, which can be

observed in the gamma attenuation or in the magnetic susceptibility data. The transition core (Fig. 7) highlights the MSCL

ability to detect sharp changes in frozen bulk density/volumetric ice content related to cryostratigraphic changes. At 8 cm depth the transition core's sharp change is observed in the MSCL results as a rapid, sustained increase in frozen bulk density from 0.9 g/cm$^3$ to ~1.6 g/cm$^3$. At the same time, the cuboid samples in this area fell on either side of the transition resulting in a sharp increase in frozen bulk density between the 6-8 cm and 8-10 cm cuboids. Whereas in the diamict and ice-rich silt cores, this may not always be the case and sharp changes in density may be smoothed by the lower resolution cuboid sampling. The gradual increase observed in the gamma attenuation density results within the transition core can be related to the diameter of the gamma beam relative to the acquisition resolution. In other words, a measurement was collected every 5 mm, but the diameter of the area covered by every measurement was ~10 mm resulting in averaging between neighbouring data points. Therefore, if a sharp local density high (ice-poor sediment layer) or density low (ice layer) occupies less than 10 mm of the vertical profile the density contrast across the boundary will be slightly muted. A good example of this is the ice layer in the ice-rich silt core at ~6 cm depth. This ice layer is picked up by the gamma attenuation sensor, although the lowest density recorded across the ice layer is ~1.1 g/cm$^3$, suggesting that the surrounding sediment was still influencing the attenuation of the gamma while in the center of the ice layer (which would have an expected density of ~0.9 g/cm$^3$). By contrast the destructive cuboid method shows no record of the ice layer in terms of a relative density low.

### 4.3 Impact of Inaccurate Soil Dry Bulk Density Measurements on Non-destructive VIC Estimates.

Estimates of volumetric ice content are dependent on accurate, spatially resolved measurements of frozen bulk density and an estimation of soil dry bulk density. For example, between 8-12 cm depth, the transition core differs from the cuboid volumetric ice content measurements and the MSCL equation results. This is caused by a local spike in the soil dry bulk density due to a sand dominated layer. This was confirmed visually using the MSCL line-scan image of the core and can be observed in the frozen bulk density measurements from the cuboid data near these depths (Fig. 7). The magnetic susceptibility also records a local high indicating the presence of minerals with high magnetization potential amongst the coarser sediments. This scale of stratigraphic variation can be picked up with visual inspections of the MSCL core images and complementary datasets while maintaining a non-destructive approach to estimating volumetric ice content. It should be noted that the volumetric ice content data are calculated based on average estimated soil dry bulk density and therefore represent an estimation rather than a measurement. Additionally, this study found that the non-destructive method saw a decrease in accuracy in heterogenous samples. This is likely related to sample resolution contrast between the destructive and non-destructive methods. Further, mixed sediments with variable organic content make dry soil bulk density estimation difficult resulting in a decrease in the accuracy of the associated volumetric ice content estimations. Nonetheless, this non-destructive method for estimating volumetric ice content is in close agreement with the destructive cuboid results and only requires an estimate of the average soil dry bulk density and assumption of air content.

**4.4 Future Implications for Application of MSCL on Permafrost Cores.**

The overall agreement between the cuboid and MSCL results shown in Figures 11, 12, 13, and 14 illustrate the MSCL's ability to be a reliable and accurate non-destructive method for collecting frozen bulk density measurements from a wide variety of frozen materials. In addition to the higher resolution the MSCL provides, relative to most destructive methods, it also can collect data from tens of meters of core in a single day (depending on sampling resolution) while leaving the sample uncompromised for future investigations. This represents a substantial time saving in comparison with traditional destructive methods.

A benefit of the MSCL for permafrost applications is the development of digital archives detailing fundamental physical properties of previously collected permafrost cores. Permafrost cores are collected typically from remote locations at great expense and are usually completely consumed in the analysis of physical properties. A digital archive containing high quality data including volumetric ice content, frozen bulk density, magnetic susceptibility, and high-resolution imagery provides an important resource for future permafrost investigations. This includes the retention of information from permafrost affected sites which may experience complete thaw in the coming years.

**5 Conclusions**

Ice content is a critical variable when considering the impact of thawing permafrost in the context of climate change. There is a clear need to develop rapid, non-destructive methods to quantify volumetric ice content from a range of depositional settings to evaluate landscape responses to warming temperatures. Relative to previous destructive methods, MSCL provides a fast, non-destructive method for collecting high resolution frozen bulk density measurements and volumetric ice content estimations from permafrost cores. The development of a MSCL permafrost protocol and redesigned frozen calibration piece, presented in this study, allows for accurate frozen bulk density measurement (RMSE = 0.067 $g/cm^3$) to be collected at resolutions approaching 0.5 cm, while measuring cores at ~2-3 cm per minute. We demonstrate a rapid approach to estimate the volumetric ice content using a combination of the frozen bulk density from the gamma density measurements and measured soil dry bulk density. The results compare well with traditional destructive methods demonstrating the potential of this method (RMSE = 6.7%). The continued development and implementation of the MSCL within the permafrost community would greatly aid in the transition to a formal standardized method for collecting physical properties from permafrost cores and ultimately robust permafrost borehole digital archives.

**Author contribution**

JP, DF, and AM planned the project; JP, AA, and CB collected the samples; JP, AM, and JH developed the methods; JP, AM, and JH performed the measurements; JP, MH, and JH analyzed the data; JP, AM, and MH wrote the manuscript draft; JP, MH, AM, JH, AA, CB, and DF reviewed and edited the manuscript.

**Competing interests**

The authors declare that they have no conflict of interest.

**Acknowledgements**

The authors would like to thank Evan Francis, Emma Braun, and Jack Bennett who helped with gravimetric measurements in earlier experiments. We would also like to thank Geotek for the support and constructive feedback they provided throughout the project.

**Financial support**

This research was supported by the NSERC funded Permafrost Partnership Network for Canada (PermafrostNet) and NSERC Discovery grant to Duane Froese. Laboratory infrastructure for the Permafrost Archives Laboratory was funded by Canadian Foundation for Innovation, Government of Alberta, and University of Alberta.

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

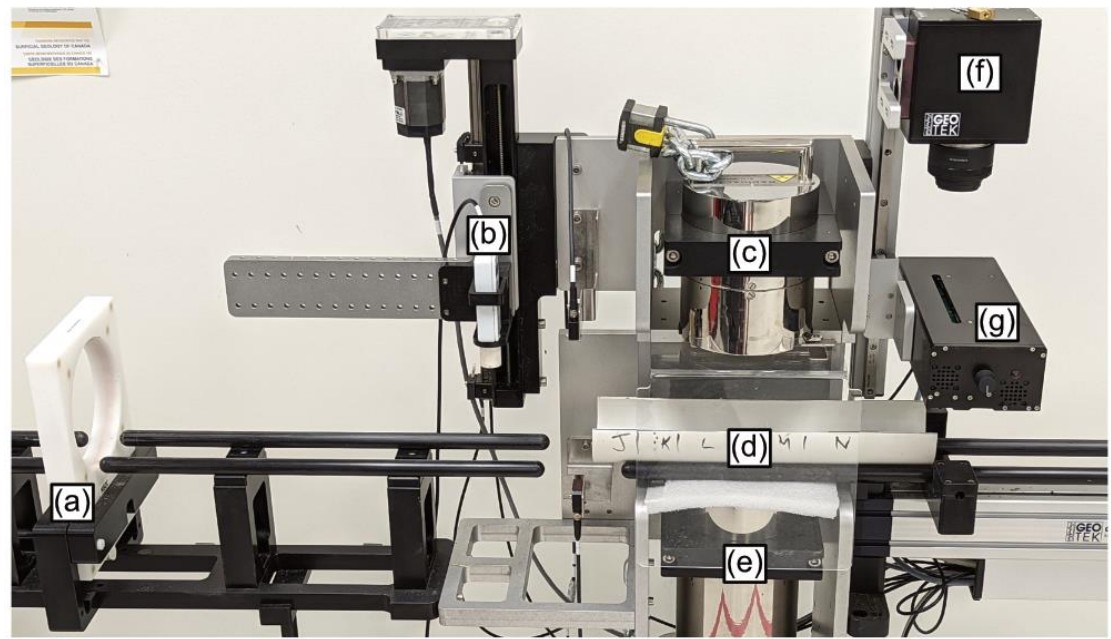

**Figure 1: The PACS Laboratory MSCL is equipped with the following components: (a) loop magnetic susceptibility sensor, (b) point magnetic susceptibility sensor, (c) Caesium-137 ($^{137}$Cs) gamma ray source, (d) core boat (40 cm in length) (e) gamma detector, (f) imaging camera box, and (g) camera light box.**

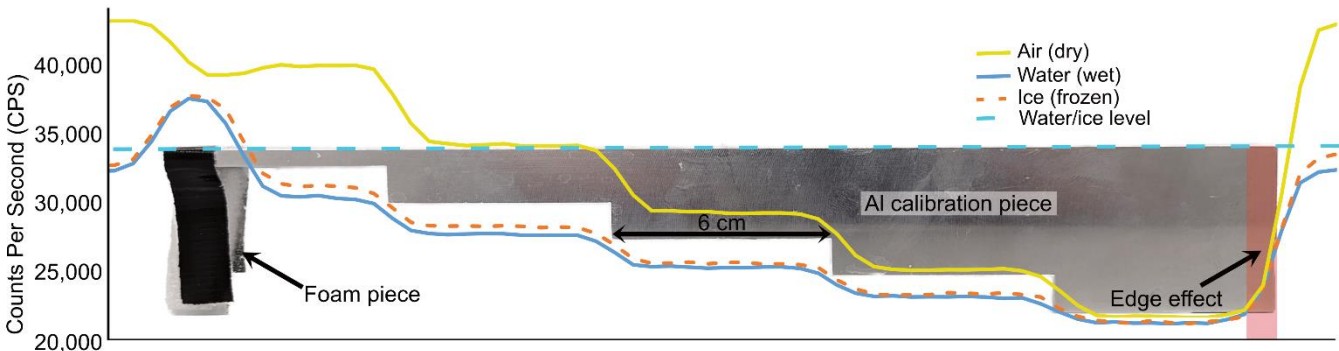

**Figure 2: Comparison of the wet, frozen (ice) and dry calibration of gamma density.**

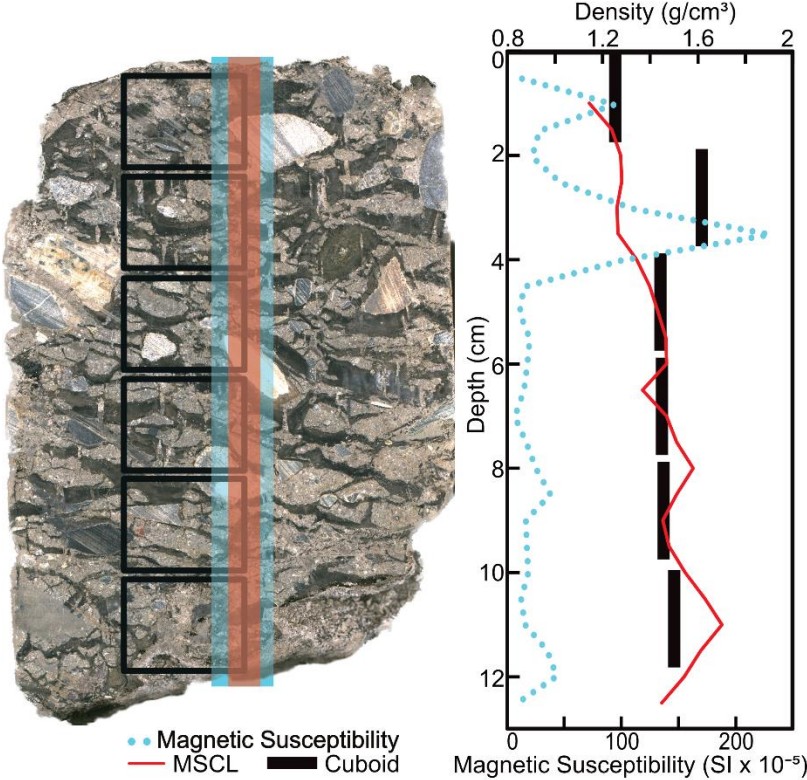

**Figure 3: Image of a core highlighting the destructive subsample (cuboid) locations (in black) relative to the non-destructive data paths of the MSCL transects.**

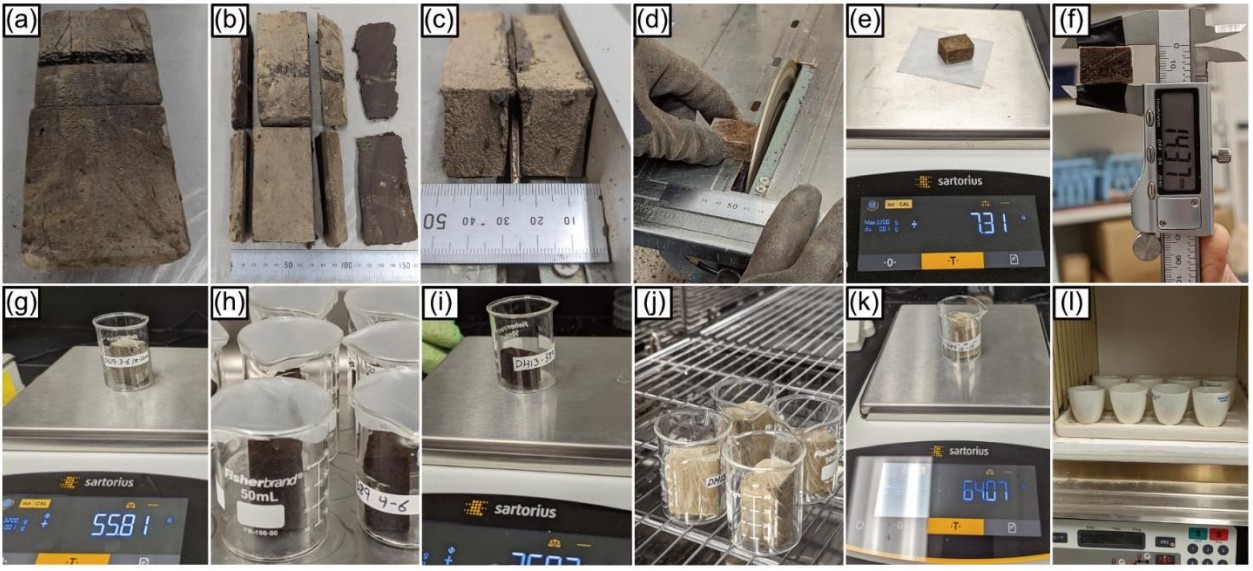

**Figure 4: Images of the cuboid method as implemented for this study.**

**Table 1: Sampling location and physical properties of cores analyzed in the study. Soil dry bulk densities were measured via the cuboid method. * *Peat/organic soil dry bulk densities sourced from Kazemian et al. (2011) and Motorin et al. (2017).***

| Core ID | Length (cm) | Classification/Properties | Collection Location | Cuboid measured Soil Dry Bulk Density (g/cm$^3$) |
|---|---|---|---|---|
| BH19-204 | 21 | Ice-wedge | Alaska HWY, Yukon, Canada | N/A |
| BH18-211 | 23 | Ice-rich silt | Alaska HWY, Yukon, Canada | 2.02 (n=20) |
| BH12F-138 | 16 | Transition between ice-rich silty peat and ice-poor inorganic silt | Alaska HWY, Yukon, Canada | 1.75 (top; from literature*) 2.17 (bottom; n=8) |
| BS19-3-6 | 19 | Diamicton | Dempster HWY, Yukon, Canada | 2.44 (n=12) |
| BH20B-337 | 20 | Ice-poor silt | Alaska HWY, Yukon, Canada | 2.67 (n=18) |
| DH13-589 | 26 | Ice-rich homogenous peat. | Dempster HWY, Yukon, Canada | 1.55 (from literature*) |


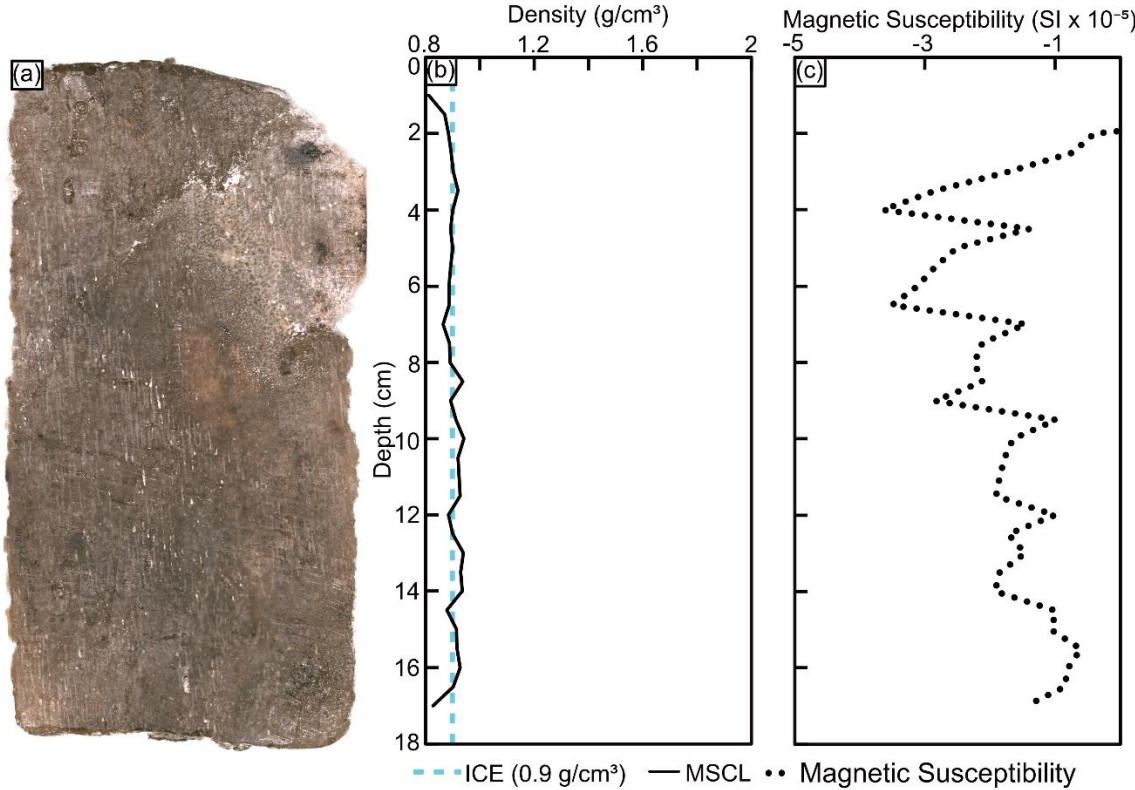

**Figure 5: (a) MSCL image of the ice-wedge core; (b) gamma attenuation density; (c) magnetic susceptibility.**

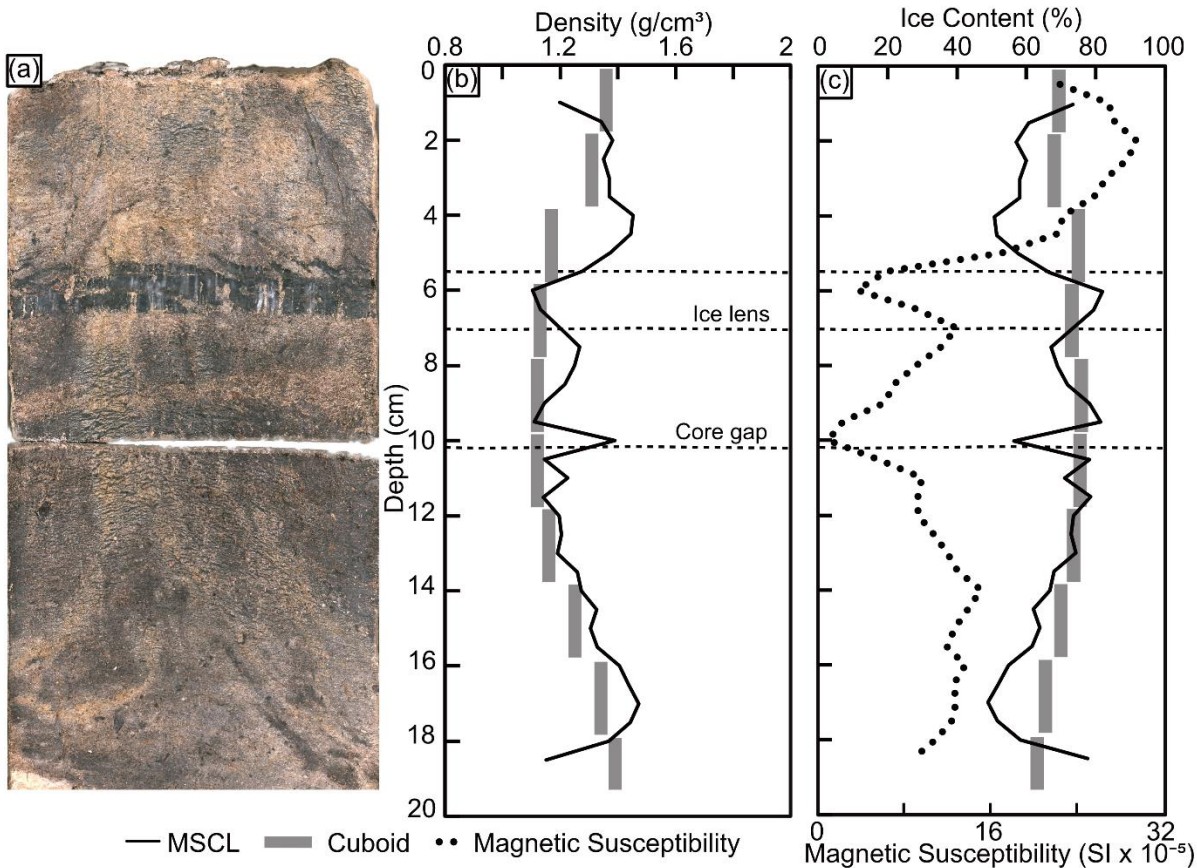

**Figure 6: (a) MSCL image of ice-rich silt core; (b) frozen bulk density comparison between gamma attenuation and cuboid data; (c) estimated volumetric ice content from frozen bulk density and cuboid, and magnetic susceptibility data.**

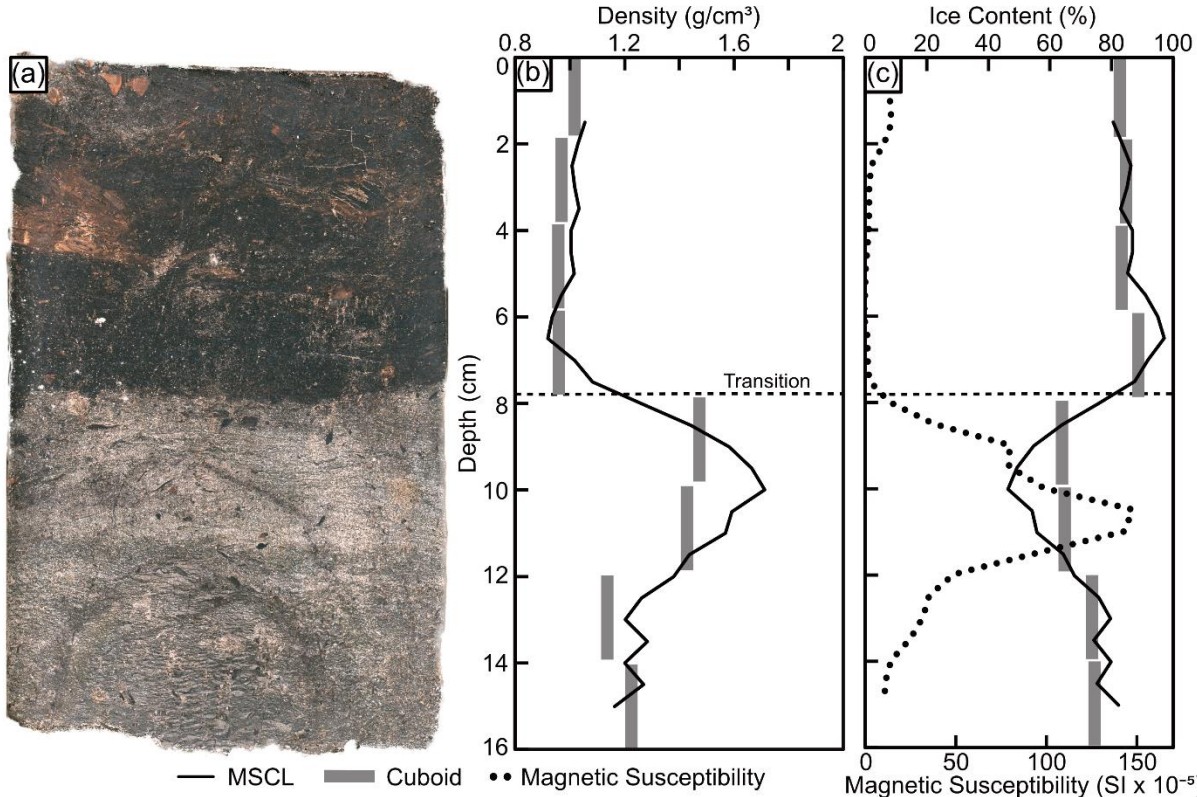

**Figure 7: (a) MSCL image of transition core; (b) frozen bulk density comparison between gamma attenuation and cuboid data; (c) estimated volumetric ice content from gamma attenuation frozen bulk density, cuboid ice content and magnetic susceptibility data.**

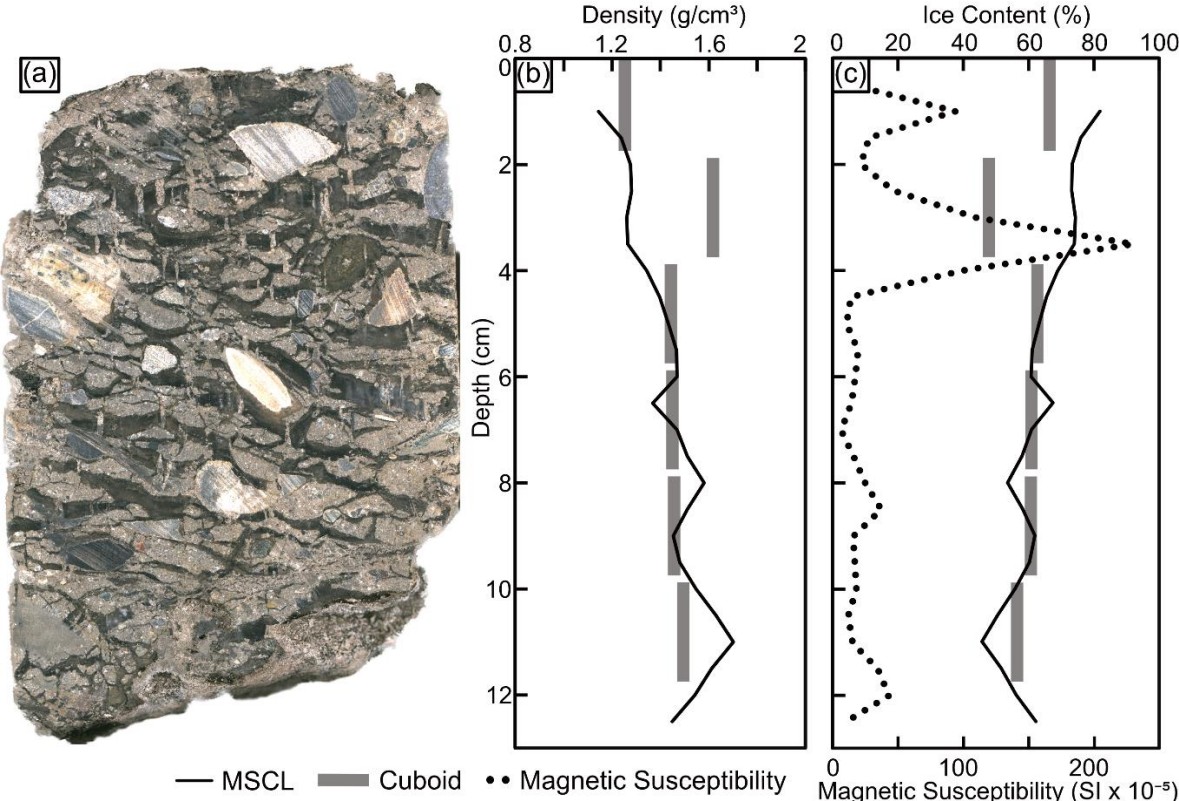

**Figure 8: (a) MSCL image of diamicton core; (b) frozen bulk density comparison between gamma attenuation and cuboid data; (c) estimated volumetric ice content from gamma attenuation frozen bulk density, cuboid ice content and magnetic susceptibility data.**

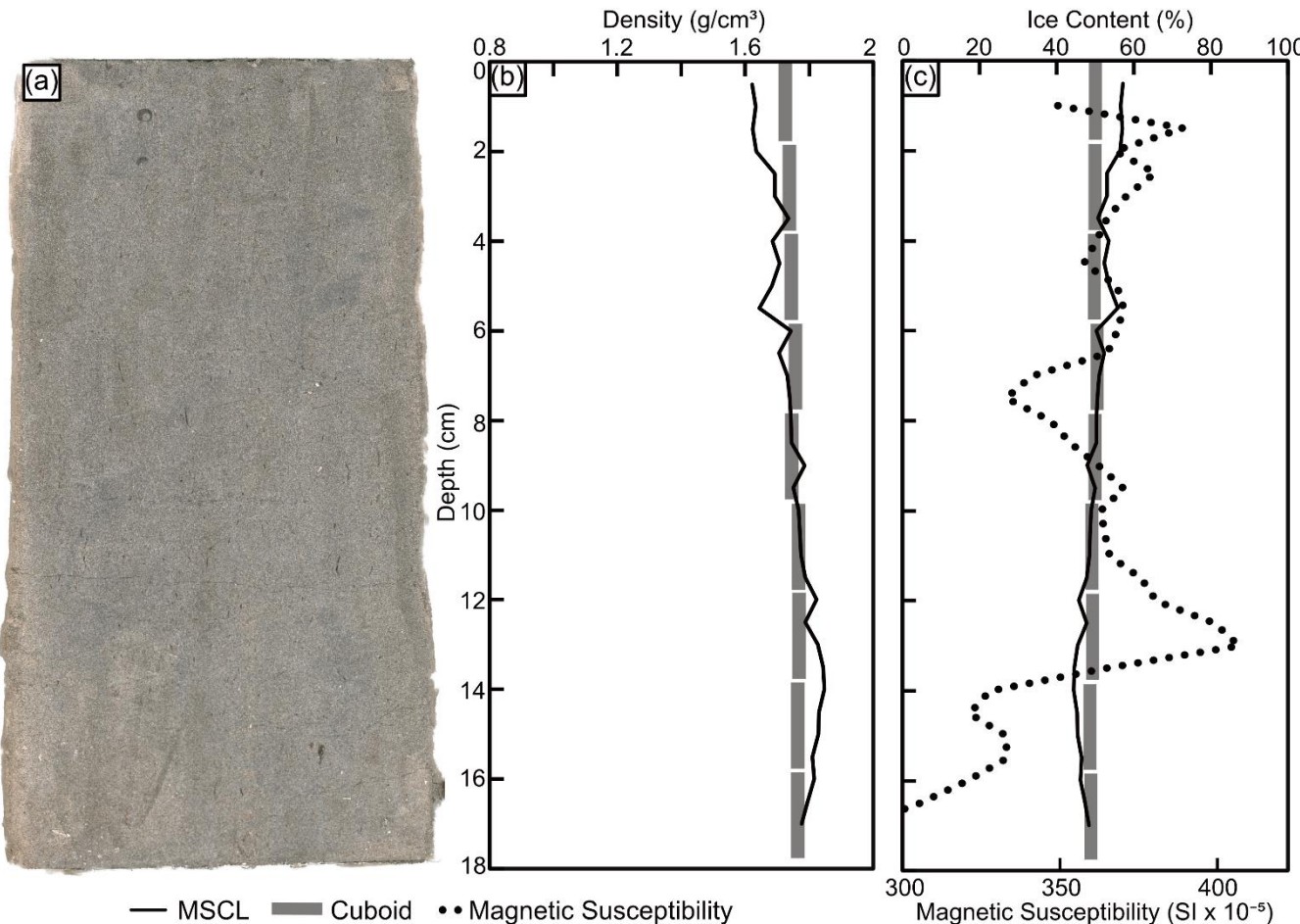

**Figure 9: (a)** MSCL image of ice-poor silt core; **(b)** frozen bulk density comparison between gamma attenuation and cuboid data; **(c)** estimated volumetric ice content from gamma attenuation frozen bulk density, cuboid ice content and magnetic susceptibility data.

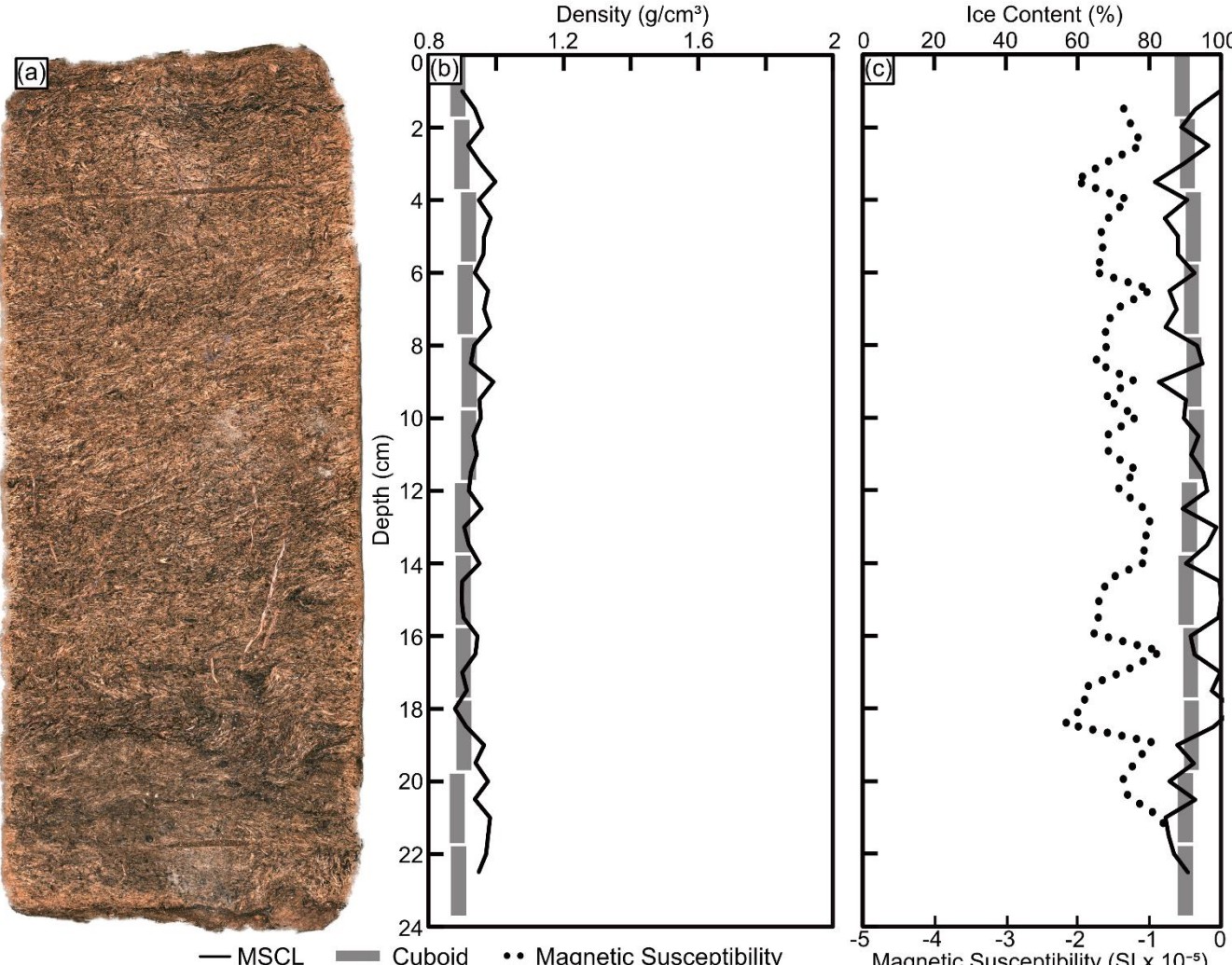

**Figure 10: (a) MSCL image of peat core; (b) frozen bulk density comparison between gamma attenuation and cuboid data; (c) estimated volumetric ice content from gamma attenuation frozen bulk density, cuboid ice content and magnetic susceptibility data.**

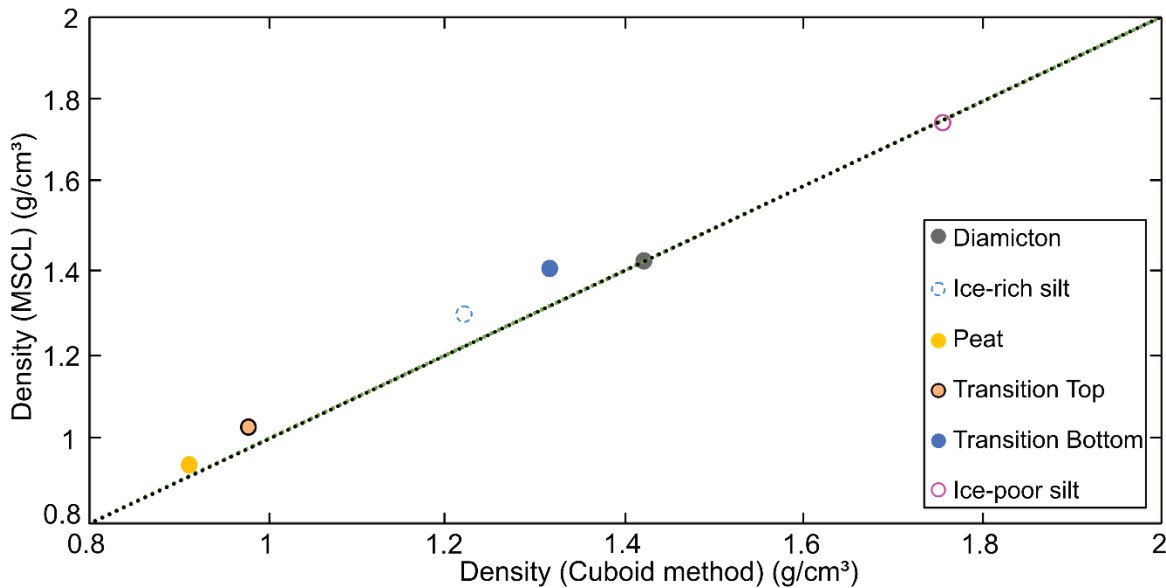


**Figure 11: Comparison between core-based averaged frozen bulk density data for the cuboid method and the MSCL. The black dotted line represents the regression line between the destructive cuboid method and the non-destructive MSCL method.**

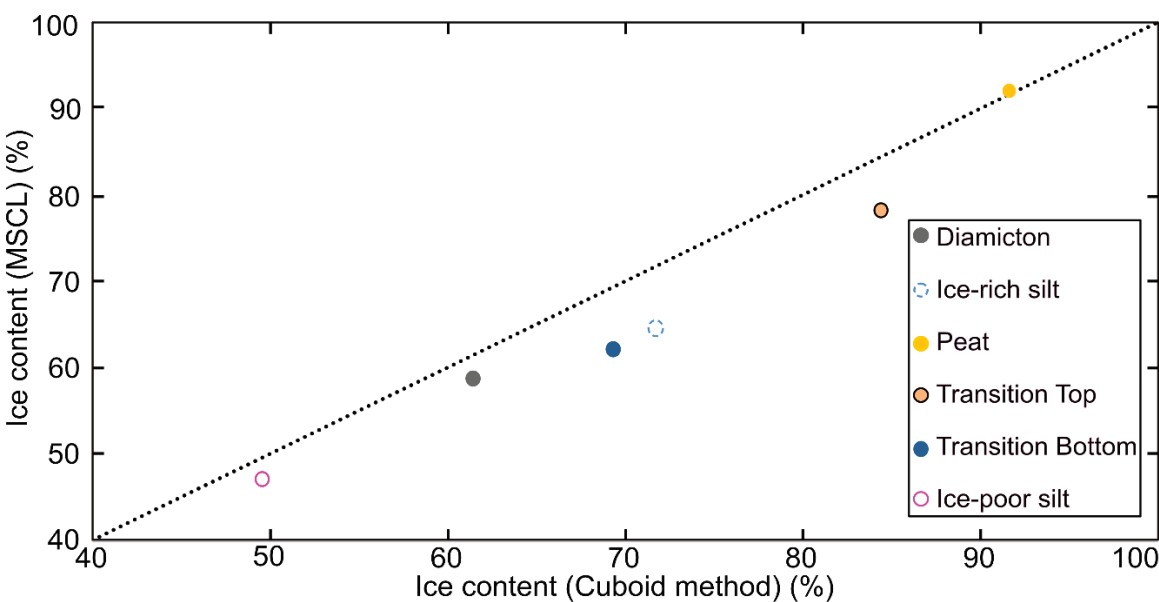

**Figure 12: Comparison between core-based averaged volumetric ice content results from the cuboid and estimates from the MSCL.**
**The black dotted line represents the regression line between the destructive cuboid method and the non-destructive MSCL method.**

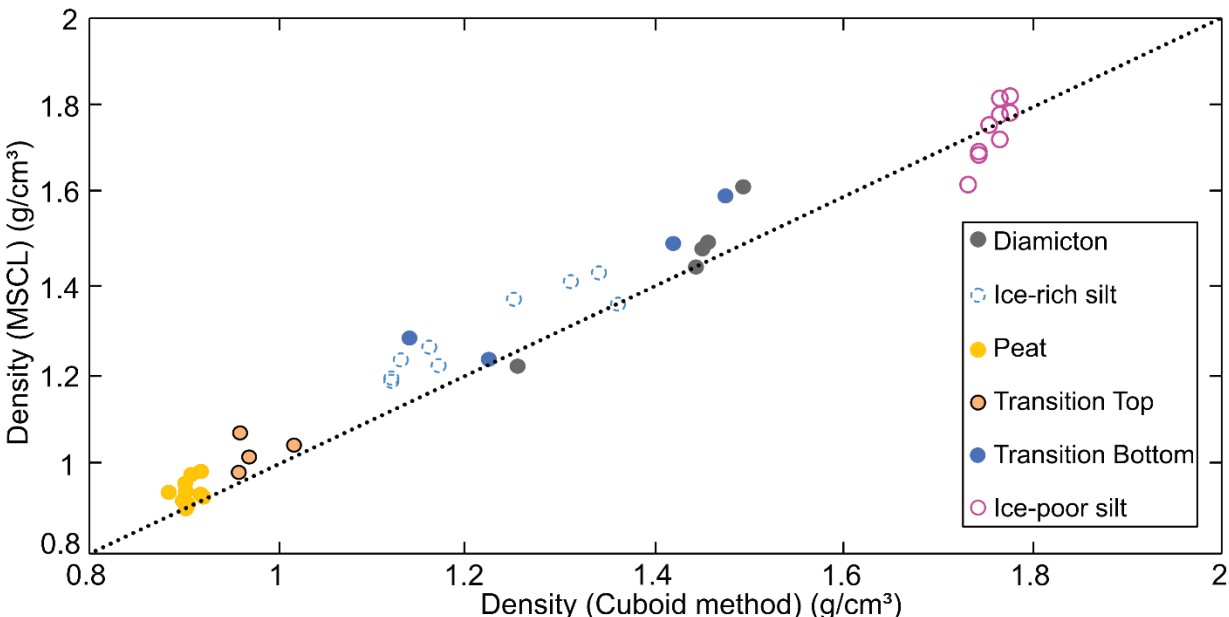

**Figure 13: Comparison between discrete sample based frozen bulk density data for the cuboid method and the MSCL. The black dotted line represents the regression line between the destructive cuboid method and the non-destructive MSCL method.**

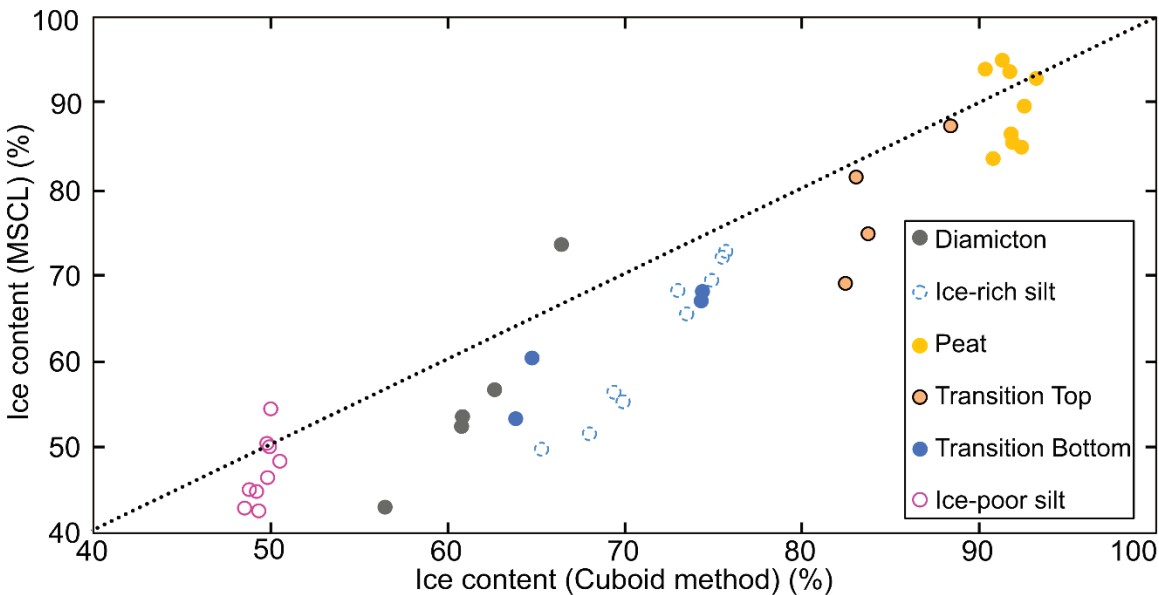

**Figure 14: Comparison between discrete sample based volumetric ice content results from the cuboid and estimates from the MSCL. The black dotted line represents the regression line between the destructive cuboid method and the non-destructive MSCL method.**

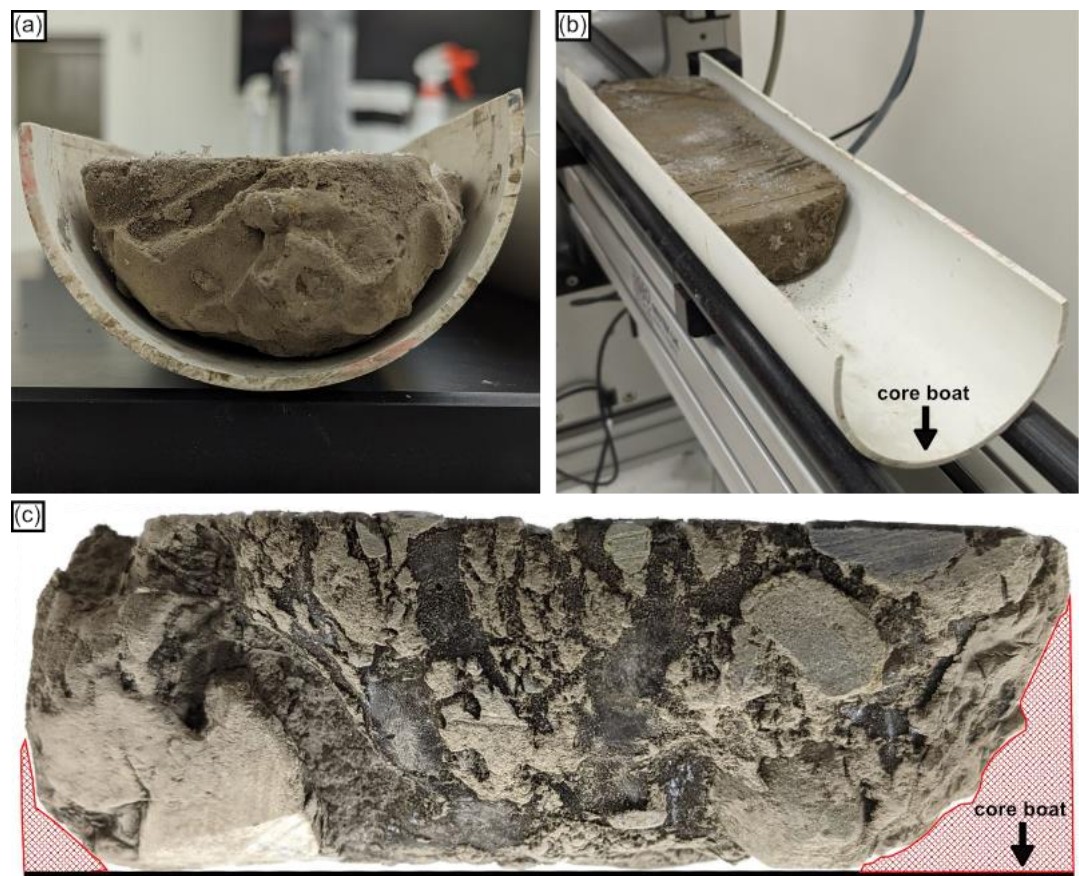

**Figure 15: The MSCL core boat and core positioning examples; (a) down core view of a core in the core boat with minimal air gaps; (b) photo of core in the core boat on the MSCL track; (c) diamicton core (BS19-3-6) side profile to highlight the thickness variation not visible to the MSCL laser.**
