# Peer review of "Non-destructive multi-sensor core logging allows rapid imaging and estimation of frozen bulk density and volumetric ice content in permafrost cores"

_EGUsphere, 2023_

## Author Comment (AC2)

This study presents the application of a commercially available device named "Non-destructive multi-sensor core logging" to measure wet bulk density and estimate ice content in permafrost cores. I like the general concept of the paper, namely the evaluation of a device to improve the estimation of soil physical properties in a non-destructive way and faster than conventional destructive methods. I acknowledge the authors for their lab studies and the value of performing these. However, I have several major concerns on the message that is provided (in the abstract and the manuscript), as well as the content of the paper. Here below are my suggestions that may help make the paper stronger and less prone to miss-interpretation of the results by the reader.

Major comments;

- o In the abstract, the authors claim (in two separate sentences) that this approach enables "strong agreement" for the estimation of bulk density, as well as ice-content. I think that claiming this for the ice-content without providing a quantitative value and explaining the limitations (incl., the limitation linked to the assumption of known dry bulk density and absence of air (not mentioned in the abstract) is very misleading. I suggest the author to improve the abstract and some parts of the manuscript to provide some more quantitative values of accuracy and add sufficient details to avoid a miss-interpretation of the results (with regard to the method applicability and limitations).

Response:

We appreciate the comment and see how this wording could be mis-leading. We have added text to clarify the sources of error/ limitations both in the abstract and main text.

Action:

We have adjusted the abstract to include the sources of error associated with soil dry bulk density estimates and air content. We have added the RMSE results to the abstract, text and conclusions for both data sets (Lines 18-23).

" MSCL frozen bulk density data show strong agreement with destructive analyses based on discrete sample measurements**, RMSE = 0.067 g/cm³. Frozen bulk** density data from the gamma attenuation, along with soil **dry bulk** density measurements for different sediment types, are used to estimate volumetric ice content.  **This approach does require an estimation of the soil dry bulk density and assumption of air content. However,**

**the averaged results for this method show agreement with an RMSE = 6.7%, illustrating** MSCL can provide  non-destructive estimates of **volumetric** ice contents and provide a digital archive of permafrost cores for future applications."

We discuss the RMSE results in greater detail with respect to the different sample types in the discussion section (lines 361-364).

**"Additionally, this study found that the non-destructive method saw a decrease in accuracy in heterogenous samples. This is likely related to sample resolution contrast between the destructive and non-destructive methods. Further, mixed sediments with variable organic content make dry soil bulk density estimation difficult resulting in a decrease in the accuracy of the associated volumetric ice content estimations.** Nonetheless, this non-destructive method for estimating volumetric ice content is in close agreement with the **destructive** cuboid results and only requires an estimate of the average soil **dry bulk** density**."**

We also explain in more detail the assumed 2% air content for all cores apart from the peat core and provide further explanation on the peat core air content assumptions in lines 204-210. We have also updated table 1 to reflect the correct soil dry bulk densities as the previous values were incorrect.

**"The largest sources of uncertainty using this approach are the estimated soil dry bulk density value and assumed 2% air content for all cores apart from the peat core and top of the transition core. Since organic content is the most important variable controlling both the overall soil dry bulk density and sample air content, we used an average soil dry bulk density for peat collected from Kazemian et al. (2011) and Motorin et al. (2017) of 1.55 g/cm3 for the peat core and 1.75 g/cm3 for the top of the transition core as it has lower organic content and higher mineral content. When we compared these values with the cuboid method VIC, these soil dry bulk densities relate to an average air content of up to ~6%."**

- o Figure 11 and 12 show a comparison of the destructive and non-destructive methods, including a correlation coefficient and a regression line. It does not make sense to me. The goal should be to assess the misfit between the two datasets using statistical metrics (e.g., the RMS error).

Response

Firstly, it should be noted that in our first version of the bulk density plot (Figure 11) the incorrect data were used for the ice-poor silt cores cuboid (destructive) results. This was caught by observing the results for the same core displayed in Figure 9 which did

not match the ones observed in Figure 11 (see below). We have corrected this error as seen in the new Figure 11 below. We appreciate this comment from the reviewer and the strengthening of the paper with the inclusion of statistical comparison.

[Figure]

Action:

We have removed the plots showing the complete data sets with associated $R^2$ values and instead added graphs of the averaged results from each core/sediment type for both volumetric ice content and frozen bulk density. This graph contains a line with a 1:1 slope showing perfect agreement between the two approaches. These figures provide a visual aid to the RMSE results. The following text was added to lines 273-279 to state the associated RMSE values. We have also separated the Transition core into "Transition Top" and "Transition Bottom" as they represent different material types.

**"The root mean square error (RMSE) has been calculated for each discrete sample comparison and averaged for each whole core. This statistical metric effectively illustrates the ability of the non-destructive method to estimate similar values to destructive methods.** Figure 11 shows strong agreement between the methods **with the individual core-based average RMSE values as follows: ice-rich silt RMSE = 0.085 g/cm$^3$ , silty peat (top of transition core) RMSE = 0.062 g/cm3, sandy silt (bottom of transition core) RMSE = 0.103 g/cm$^3$, diamicton RMSE = 0.064 g/cm$^3$, ice-poor silt RMSE = 0.051 g/cm$^3$, and ice-rich peat RMSE = 0.038 g/cm$^3$, and an overall average RMSE of 0.067 g/cm$^3$,** illustrating the reliability...**"**

[Figure]

**Figure 11: Comparison between core-based averaged frozen bulk density data for the cuboid method and the MSCL. The black dotted line represents the regression line between the destructive cuboid method and the non-destructive MSCL method.**

The following text was added to lines 288-292 to state the associated RMSE values.

**"**Overall, the gamma attenuation and cuboid **volumetric** ice content results show **good** agreement **with a RMSE of 6.7%** and demonstrate the potential for systematic and reliable estimation of **volumetric** ice content of permafrost cores non-destructively **(Figs. 13 and 14). The individual core-based average RMSE values are as follows: ice-rich silt RMSE = 8.4%, silty peat (top of transition core) RMSE = 8.1%, sandy silt (bottom of transition core) RMSE = 7.5%, diamicton RMSE = 8.2%, ice-poor silt RMSE = 4.1%, and ice-rich peat RMSE = 3.9%."**

[Figure]

**Figure 12: Comparison between core-based averaged volumetric ice content results from the cuboid and estimates from the MSCL. The black dotted line represents the regression line between the destructive cuboid method and the non-destructive MSCL method.**

- o I think it would be valuable to assess how the method captures the variability (in wet bulk density and ice-content) in each core independently. Statistically, when comparing cores from different regions with a wider range of soil properties, it is clear that a relationship can look good (like the one presented in figure 12). However, if the goal is to limit destructive sampling at numerous locations along a core, then it should show the ability to do so, and thus a misfit should be calculated for each core. It could also be related to the uncertainty that may be linked to the used dry bulk density. My criticism is not toward the data itself (great to publish data assessing the accuracy of a tool and discuss potential improvements), but the way the authors present the results.

Response:

This suggestion also adds to the paper by highlighting the weakness in the contrast of the sample resolution between the two methods and the difficulty of working with heterogenous materials comparing multiple methods.

Action:

We have added plots showing the entire individual sample comparison for each core and the following lines to the discussion (lines 292-300).

**"Figure 13 is a comparison plot between the destructive and non-destructive results for bulk density, displaying all individual sample results. Overall, the non-destructive method shows the ability to accurately recreate the results of the cuboid method despite the contrasting sample resolution and location. Figure 14 shows that increased core heterogeneity results in increased RMSE. The ice-poor silt and peat cores which are the most homogenous in terms of both ice content and bulk density display the lowest RMSE. The higher RMSE values seen in the heterogenous samples can be related to the difference in sample resolution and location between the cuboid method and MSCL method. Additionally, the core shape issue with the MSCL thickness laser had a compounding impact on the volumetric ice content data. This issue has since been resolved further reducing the sources of error for this MSCL method."**

[Figure]

**Figure 13: Comparison between the discrete sample based frozen bulk density data for the cuboid method and the MSCL. The black dotted line represents the regression line between the destructive cuboid method and the non-destructive MSCL method.**

[Figure]

**Figure 14: Comparison between the discrete sample based volumetric ice content results from the cuboid and estimates from the MSCL. The black dotted line represents the regression line between the destructive cuboid method and the non-destructive MSCL method.**

    ○   There is no discussion on the differences in magnetic susceptibility across the various samples. Such a discussion could give more strength to the paper.

Response:

We recognize that magnetic susceptibility was only lightly touched on in this paper as the focus was more on comparing destructive results to non-destructive alternatives.

Action:

We have added a short section to address the general observations on the magnetic results in lines 311-313.

**"This same clast also caused a local magnetic susceptibility peak in this core, marking one of the few exceptions to the otherwise inverse relation between magnetic susceptibility and ice content observed in the cores."**

Minor comments:

- o The authors describe this approach as "rapid". I think it is misleading (in the abstract at least). Please consider comparing the time it takes with other non-destructive methods such as a X-ray computed tomography for example (which can also deliver the wet bulk density if calibrated). Or at least please provide a value for the acquisition speed to be more quantitative.

Response/Action:

Line 13 has been adjusted to include the average acquisition time as stated in line 354 of the conclusions.

**"In this study, multi-sensor core logging (MSCL) is used to provide a rapid (~2-3 cm core depth per minute),** high-resolution, non-destructive method...**"**

- o This study uses a tool that already exists and has been applied in many environments. I assume it has been compared with other methods (destructive and non-destructive ones). In this study, there is very little discussion of application to other environments where assessing wet bulk density is likely a goal too. Also, there is no discussion of how the accuracy of this method compares to other non-destructive methods in permafrost or other environments (e.g., CT-scans, NMR, etc). Consider improving the discussion of these topics.

Response:

Lines 43-46 reference how the MSCL was used historically.

"The MSCL is well established in its application for marine sediments (Weber et al., 1997; Gunn and Best, 1998), landslide assessments (Hunt et al., 2011; Vardy et al., 2012), contaminated sediments (Kuras et al., 2016), and environmental studies of lake sediments (Smol et al., 2001; Fortin et al., 2013). However, to our knowledge, the application of MSCL on frozen materials has not been developed."

Action:

We have added some lines to briefly address the MSCL method and its specific application and accuracy relative to CT and NMR in lines 211-215.

**"Although this method requires the estimation of both the dry soil bulk density and sample air content it represents an important non-destructive tool for the permafrost core-based sciences. We recognize the MSCL can only analyze core style materials, however we present this method as an alternative that represents both a cost and time savings relative to other methods for non-destructive extraction of volumetric ice content such as nuclear magnetic resonance and computed tomography scanning."**

- Many people in the soil science community consider bulk density as equal to dry bulk density and use "wet bulk density" for the in-situ density. I suggest the author to use "wet bulk density" to make it clear to the reader. Also, I suggest replacing "soil density" (in the abstract) with soil dry bulk density. In the manuscript it is referred as soil particle density and then soil specific density (see line 183 and 190). I think the authors mean soil dry bulk density here. At least please make sure the wording is consistent over the manuscript.

Response/Action:

For greater clarification, we have changed "soil density" to "soil dry bulk density" and "bulk density" to "frozen bulk density" throughout the manuscript.

- Consider adding a discussion on the impact of organic matter content on the method accuracy.

Response:

We have an ongoing project focused on gaining a better understanding of the impacts organic content has on ice content, dry soil bulk density, and air content. This is beyond the scope of the present study.

Action:

Now directly discussed in section 4.3 lines 363-366.

**"Additionally, this study found that the non-destructive method saw a decrease in accuracy in heterogenous samples. This is likely directly related to the sample resolution contrast between the destructive and non-destructive methods. Furthermore, mixed sediments with variable organic content made dry soil bulk density estimation difficult resulting in a decrease in the accuracy of the associated volumetric ice content."**

---

## Author Comment (AC3)

- J. Pumple and co-authors present a novel approach to estimate bulk density and volumetric ice content on permafrost cores. The study has been carried out thoroughly, the paper is very well written and of interest to the readers of the Cryosphere. The method is still in its early stages for this application but those are promising. Below are some minor comments and suggestions, which the authors may want to address prior to final submission:

Firstly, thank you for your time and effort in reviewing our work.

- Title: I recommend that the title be changed to "Non-destructive multi-sensor core logging allows rapid imaging, estimation of bulk density and volumetric ice content in permafrost cores" as the method is an estimation for both parameters.

Response:

We agree that the volumetric ice content is an estimation. Following this comment, we will change bulk density to also be an estimation. This was a discussion during the early stages of the project. We went with measurement given the close agreement with measured bulk density (destructive) but agree we are in fact measuring gamma ray attenuation and estimating bulk density from those values.

Action:

 The title has been changed to the following: "**Non-destructive multi-sensor core logging allows rapid imaging and estimation of bulk density and volumetric ice content in permafrost cores**".

- In general, the authors are encouraged to always use *volumetric ice content* and not just *ice content*

Response:

Agreed and changed.

Action:

We have switched all instances of "ice content" to "volumetric ice content" where applicable.

- In the introduction it's also worth noting that not only the recovery of the samples is expensive and complex, but also the storage on site and the transport, specifically if the thermal state of the sample should be protected.

Response:

Agreed and changed.

Action:

We have adjusted line 27 to include transportation and storage; "Despite the considerable cost involved in the recovery, **transportation and storage** of permafrost cores, most methods are destructive and rarely preserve physical or digital archives for future work."

- Add a reference to BNQ 2501-500 in the introduction regarding ice content

Response:

This is a great resource for geotechnical work and sampling and w have added the reference.

Action:

We have added this reference to lines 29 and 417.

- The paper does not mention salinity. However, in polar region, the determination of the salinity of permafrost samples is important as it impacts unfrozen water content and freezing point depression, hence the soil freezing characteristic curve.

Response:

We recognize the importance of salinity in permafrost but did not address it specifically in this study.  We have added reference to it in the main text.

Action:

Following this comment, we have added a short statement to address the absence of salinity from this study (lines 186-190):

**" The cuboid method provides an opportunity to collect pH and conductivity measurements from ice rich samples following the thawing stage; however, for this study these data were not collected. We recognize the importance of salinity in thaw sensitive permafrost regions however given the analytical constraints, thermal stability was top priority during our analysis. The hope is to consider free water and salinity in future studies using alternative non-destructive methods (e.g., Roustaei et al., 2022)."**

- It is understood that the sample are stored at -25°C and the test being carried out at ~-12°C. In the ground, permafrost temperatures are much warmer and often the unfrozen water content is a critical parameter. It is also important to recognize that many soils have a freezing hysteresis, i.e. unfrozen water contents are different when thawing compared to freezing. How was the change in the soil structure, e.g. in response to freezing of unfrozen water when the sample was taken from the field and later stored, considered? Also, in section 2.1.3 the authors mention (line 140) that "… these electrical currents are likely to be altered by the differing abundance of ice and water …". However, it is questionable how much unfrozen water is still present in the sample for the conditions the samples were tested at.

Response:

We agree that the initial conditions of the permafrost are not being represented in this study but that was beyond the scope of this study—that focuses on measurement and estimation of physical properties. We have done tests related to temperature dependent physical properties (e.g. Roustaei et al., 2022) but that is beyond this project scope. Here, we focus on robust acquisition conditions and measurement and comparison to high resolution destructive analyses. We have added some lines to make this point clearer.

Action:

We address the concern about initial ground temperature of the core's vs lab tested conditions (lines 96-98):

"**The data collected in this study are under colder temperatures than ambient field conditions.  Future development will focus on designing of a chilling boat for the samples to maintain samples at much warmer temperatures (-0.5 – 5 C°) during measurement.**"

We address the comment about unfrozen water content at stated acquisition temperatures (lines 149-150):

"**We recognize that unfrozen water content will be minimal at temperatures below -5 C° and so an alternative insulated core boat would be needed if the sensors temperature sensitivity could be addressed.**"

- It would be interesting to compare the ice content with ice contents derived from image analysis. Similar to Arenson et al. (2008), it should be possible to get the ice structure from the images taken, specifically on samples such as the one shown in Figure 8.

Response:

We have a related project working on this which will extract ice content data from images using machine learning to create an automated approach. However the project was in its early stages during the final preparation of this manuscript.

- With regards to the core boat challenge, i.e. the air cap between the sample and the boat, it may be worth evaluating the possibility of creating a 3D scan of the samples and use a 3D printer for the perfect core boat.

Response:

We have recently developed a core boat with a transparent or void space bottom to address the impact of uneven core surfaces. Although we do have access to 3D printers the cost, time and waste would make this approach not viable for our research.

Action:

We now make mention that the thickness issue associated with the core boat/thickness laser has been solved (lines 296-298):

"**Additionally, the core shape issue with the MSCL thickness laser (Sect. 4.1) had a compounding impact on the volumetric ice content data. This issue has since been resolved further reducing the sources of error for this MSCL method.**"

- Line 126: Check that you always use 'e.g.,'

Response/Action:

Changed.

- Line 252: check superscript for cm$^3$

Response/Action:

Changed.

- Line 356: make sure to use 'NSERC PermafrostNet' and not just 'PermafrostNet' in brackets.

Response/Action:

Changed.

Table 1: delete '.' After peat in sample DH13-589

Response/Action:

 Changed.